# Differential Colonization and Mucus Ultrastructure Visualization in Bovine Ileal and Rectal Organoid-Derived Monolayers Exposed to Enterohemorrhagic *Escherichia coli*

**DOI:** 10.3390/ijms25094914

**Published:** 2024-04-30

**Authors:** Minae Kawasaki, Yoko M. Ambrosini

**Affiliations:** Department of Veterinary Clinical Sciences, College of Veterinary Medicine, Washington State University, Pullman, WA 99164, USA; minae.kawasaki@wsu.edu

**Keywords:** adult stem cells, bovine intestine, EHEC, Enterohemorrhagic *Escherichia coli*, immune response, *in vitro* infection model, mucus, organoid-derived monolayer, scanning electron microscopy

## Abstract

Enterohemorrhagic *Escherichia coli* (EHEC) is a critical public health concern due to its role in severe gastrointestinal illnesses in humans, including hemorrhagic colitis and the life-threatening hemolytic uremic syndrome. While highly pathogenic to humans, cattle, the main reservoir for EHEC, often remain asymptomatic carriers, complicating efforts to control its spread. Our study introduces a novel method to investigate EHEC using organoid-derived monolayers from adult bovine ileum and rectum. These polarized epithelial monolayers were exposed to EHEC for four hours, allowing us to perform comparative analyses between the ileal and rectal tissues. Our findings mirrored *in vivo* observations, showing a higher colonization rate in the rectum compared with the ileum (44.0% vs. 16.5%, *p* < 0.05). Both tissues exhibited an inflammatory response with increased expression levels of *TNF-a* (*p* < 0.05) and a more pronounced increase of *IL-8* in the rectum (*p* < 0.01). Additionally, the impact of EHEC on the mucus barrier varied across these gastrointestinal regions. Innovative visualization techniques helped us study the ultrastructure of mucus, revealing a net-like mucin glycoprotein organization. While further cellular differentiation could enhance model accuracy, our research significantly deepens understanding of EHEC pathogenesis in cattle and informs strategies for the preventative measures and therapeutic interventions.

## 1. Introduction

Enterohemorrhagic *Escherichia coli* (EHEC) is a zoonotic pathogen of significant public health concern, primarily due to its association with severe gastrointestinal diseases in humans, including hemorrhagic colitis and hemolytic uremic syndrome, a potentially fatal condition [1,2]. Cattle are recognized as a principal reservoir for EHEC, with human infections typically arising from the consumption of contaminated meat and dairy products or direct contact with infected animals [1,3]. Interestingly, while EHEC infections can be life-threatening in humans, cattle, in particular adult animals, often do not exhibit clinical symptoms despite being carriers and intermittently shedding the bacteria for extended periods [1]. The rectum has been identified as the primary site for EHEC colonization in cattle, serving as a reservoir where the bacterium not only persists but also actively sheds, contributing significantly to its transmission [4,5]. The formation of attaching and effacing (A/E) lesions within this site, marked by the destruction of microvilli and the intimate attachment to enterocytes, underscores the pathogen’s capacity to breach host defenses. Such persistence of EHEC in the bovine rectum is a critical factor in the risk of foodborne outbreaks in humans. Conversely, the vulnerability of the small intestine to EHEC, particularly its ability to instigate A/E lesions, has been predominantly observed in experimental settings utilizing ileal tissue such as *in vitro* organ culture and ligated ileal loop models or among neonatal calves [6,7,8]. These findings underscore the importance of understanding the pathophysiological underpinnings of EHEC’s colonization and endurance in the bovine gastrointestinal tract, aiming to reduce the transmission threat to humans.

Current knowledge on key aspects of host-pathogen interactions in cattle, including EHEC’s natural colonization patterns as outlined above and host immune responses to EHEC infection, is largely drawn from *in vivo* observations in naturally or experimentally infected animals [4,9,10,11]. While these models can provide useful insights to improve our understanding of host–pathogen interactions, the ability to investigate the mechanisms of disease development at the molecular detail is limited; additionally, they inherently possess concerns around economical and animal welfare perspectives. A recent study described the development of bovine intestinal xenograft models in mice and their utility as an EHEC infection model as an alternative to *in vivo* calf models, significantly enhancing cost-effectiveness [12]. However, *in vitro* studies to elucidate the mechanisms regulating pathogenicity of EHEC in this species remain sparse due to the lack of effective *in vitro* models, contrasting to human counterparts where organoid-derived intestinal monolayer models have offered valuable *in vitro* tools to study host–EHEC interactions in greater detail, including EHEC’s impacts on the intestinal mucus [2,13,14]. Major advantages of organoid-derived monolayer models over three-dimensional (3D) organoid models include markedly improved accessibility to the polarized epithelial luminal interface where host–pathogen interactions take place in *in vivo* intestines. A defined cell culture surface area utilized by the monolayer system provides greater control over the host–pathogen ratio, which is more difficult to achieve with conventional or apical-out organoid models. Therefore, utilization of the monolayer system effectively overcomes limitations faced with 3D organoids as an infection model, while it retains cellular heterogeneity unique to organoids.

The mucosal response to EHEC infection, particularly the function of the mucus barrier in modulating disease severity, is a critical area of research. The mucus barrier not only physically impedes bacterial adhesion to the epithelial surface but also engages in active defense mechanisms, including the secretion of mucins in response to bacterial presence [15]. Such responses are crucial for the host’s ability to combat infection and limit disease progression, highlighting the importance of mucus ultrastructure characterization to improve our understanding of host–pathogen interactions associated with the mucus layer in the natural state. However, the lack of *in vitro* models that accurately represent the bovine intestinal environment has hindered detailed studies of host–pathogen interactions and mucus structures within the bovine gut.

Addressing this gap, our study leverages advanced adult bovine ileal and rectal organoid-derived monolayers as *in vitro* models to simulate an EHEC infection. This approach enables a comprehensive investigation into the bacterial colonization patterns, tissue-specific responses, and the efficacy of mucosal defenses against EHEC. By comparing ileal and rectal monolayers, we aim to elucidate differences in host cell responses and the impact of EHEC on epithelial integrity across these two key intestinal sites. Additionally, the study explores fixation techniques for preserving and visualizing the mucus ultrastructure, thereby providing insights into the natural mucosal barriers against EHEC adhesion and infection. This research not only enhances our understanding of EHEC pathogenesis in cattle but also contributes to the development of strategies to prevent foodborne transmission of this pathogen to humans.

## 2. Results

### 2.1. EHEC Demonstrated Preferential Colonization to Adult Bovine Rectal Organoid-Derived Monolayers over Ileal Counterparts

The ileal and rectal monolayers derived from adult bovine organoids were confirmed to be stable prior to EHEC exposure using phase contrast microscopy, transepithelial electrical resistance (TEER) measurements, and paracellular permeability (*P*_app_) assay (Appendix A). Host–pathogen interactions and host immune response to bacterial infection were evaluated using electron microscopy and RT-qPCR analyses at 4 h post infection (Figure 1a). Scanning electron microscopy (SEM) documented bacterial adherence to the apical surface of the monolayers and to the extruded mucus in both ileal and rectal monolayers (Figure 1b). The number of bacteria adhering to the epithelium was significantly greater in rectal monolayers compared with that in ileal counterparts (44.0 ± 18.9% vs. 16.6 ± 5.5%, *p* < 0.05) (Figure 1c).

Further evaluation of the monolayers with transmission electron microscopy (TEM) revealed structural disturbance and effacement of microvilli in EHEC-infected ileal and rectal monolayers, respectively, compared with uninfected controls (Figure 1d). Disorganization or loss of uniform glycocalyx structure was also noted in both ileal and rectal monolayers after infection with EHEC. Characteristic actin pedestal formation, representative of A/E lesion, was not readily observable in either ileal or rectal monolayers at 4 h.

Early immune response of epithelial cells to EHEC infection was noted as upregulation of major proinflammatory cytokine genes *TNF-a* and *IL-8*, but not *IL-6*, in both ileal and rectal monolayers relative to uninfected controls (Figure 1e). While upregulation of the *TNF-a* gene was significant in both ileal and rectal monolayers at 4 h post infection with EHEC (*p* < 0.05), that of *IL-8* was significant only in rectal monolayers (ileum: *p* = 0.09 and rectum: *p* < 0.01).

### 2.2. EHEC-Induced Actin Filament Remodeling Was More Dramatic in Rectal Monolayers than in Ileal Monolayers

The impact of EHEC on epithelial integrity was assessed with phase contrast and confocal microscopy. Overall, phase contrast microscopy analysis confirmed that both ileal and rectal monolayers remained intact at 4 h post infection with EHEC (Figure 2a). However, immunofluorescence staining against F-actin revealed actin filament remodeling and apical membrane perturbation in EHEC-infected rectal monolayers as reported previously [2] (Figure 2b). Similar changes were not readily detectable in EHEC-infected ileal monolayers. The impact of EHEC on the expression of adherens junction protein, E-cadherin, actin filament, and tight junction genes was minimal upon immunofluorescence staining (Figure 2c) and RT-qPCR analysis (Appendix A).

### 2.3. Mucus Covering Area on the Apical Surface of Ileal Monolayers Significantly Increased Following Infection with EHEC

The impact of EHEC on the mucus layer which covers the apical surface of the monolayers was assessed with immunofluorescence microscopy. *Sambucus nigra* agglutinin (SNA) staining of ileal monolayers revealed more generalized positive signals over the apical surface of both the control and the EHEC-infected monolayers, demonstrating the presence of a mucus layer (Figure 3a,b). The SNA-positive area was significantly greater in the EHEC-infected monolayers relative to the uninfected control (13.5 ± 3.6% vs. 6.1 ± 2.0%, *p* < 0.01) (Figure 3c). In contrast, SNA-positive signals detected in rectal monolayers were restricted to the area within or directly apical to mucus-producing goblet cells in both control and EHEC-infected monolayers, indicating a sparsity in the surface covering mucus layer (Figure 3a,b). The SNA-positive area was not different between the control and the EHEC-infected monolayers (0.7 ± 0.1% vs. 0.5 ± 0.1%, *p* = 0.21) (Figure 3c). RT-qPCR analysis revealed no differences in expression levels of the *MUC2* gene between the control and the EHEC-infected monolayers in both the ileum (*p* = 0.84) and the rectum (*p* = 0.31) (Appendix A).

### 2.4. Mucus Ultrastructure Is Better Visualized with Modified Fixation Protocol under SEM

Two fixation techniques, namely glutaraldehyde (GA) alone and GA mixed with paraformaldehyde and alcian blue (GA/PFA/AB), were compared for their effectiveness in preserving and visualizing mucus ultrastructure under SEM. Evaluation of control monolayers revealed that GA-fixation preserved little mucus covering the apical surface of the epithelium, allowing clear visualization of characteristic cellular structures such as microvilli (Figure 4a). Consequently, this technique effectively highlighted differing structural characteristics of ileal and rectal monolayers, i.e., more densely packed longer microvilli vs. more sparse shorter microvilli, respectively. In contrast, GA/PFA/AB-fixation preserved more mucus which uniformly covered the apical surface of monolayers, thus facilitated visualization of mucus ultrastructure (Figure 4b). SEM analysis of mucus ultrastructure revealed generally uniform net-like 3D organization of mucin glycoproteins in both ileal and rectal monolayers [16].

EHEC-infected ileal and rectal monolayers were prepared with both fixation techniques and impact of EHEC on microvilli and mucus structures were assessed with SEM. With the GA-fixation of ileal monolayers, elongated microvilli were occasionally observed among largely undisturbed uniform microvilli, whereas generalized areas of microvillous effacement was readily observable in rectal monolayers (Figure 5a). Moreover, adhering bacteria on rectal monolayers were more frequently noted with close association with distorted microvilli, i.e., elongated microvilli tightly anchoring adhering bacteria, depicting more pronounced actin remodeling, contrasting to lightly adhered bacteria with little association with surrounding microvilli observed in ileal monolayers. GA/PFA/AB-fixation revealed disturbance of mucus ultrastructure in both ileal and rectal monolayers upon infection with EHEC, with slightly more dramatic changes noted in rectal monolayers (Figure 5b). Characteristic uniform net-like organization of mucus was largely destructed, and mucin glycoproteins were extended out to form series of elongated strands and closely associated with adhering bacteria in both ileal and rectal monolayers.

## 3. Discussion

In this study, the application of bovine ileal and rectal organoid-derived monolayers as an *in vitro* model for studying EHEC colonization has been demonstrated to be feasible. Multi-modal analysis confirmed that these monolayers could replicate bacterial adherence, tissue tropism, and host immune response observed *in vivo* or *ex vivo* in bovine [4,5,12], highlighting their potential for investigating host-pathogen interactions in a controlled environment. Faithful replication of inter-segmental variations in EHEC colonization and host response to EHEC infection in *in vitro* environment marks a critical advancement toward developing effective strategies to eliminate EHEC from the intestine of bovines, which has not yet been successful to date.

Following EHEC exposure, both rectal and ileal monolayers exhibited upregulation of proinflammatory cytokine expressions, namely *TNF-a* and *IL-8*, which was consistent with previous observations using *in vivo* calf and *in vitro* cell line models, as well as in human patients infected with EHEC [9,17,18,19]. The response was more pronounced in rectal monolayers than in their ileal counterparts, likely reflecting the more extensive colonization in the rectum and a stronger stimulus for an immune response to protect the gut epithelium. This differential cellular response could be associated with varying colonization rates across different intestinal segments as observed in chicken intestinal organoids infected with *Salmonella* spp. [20].

A notable finding was the significantly higher adherence of bacteria to rectal monolayers compared with ileal monolayers, suggesting a segment-specific tropism in EHEC colonization within the gastrointestinal tract. The result is in line with previous observations utilizing various *in vivo*, *ex vivo*, and *in vitro* models spanning different species including humans, bovines, and mice [4,5,12,13,21,22], reinforcing the importance of developing and selecting segment-specific *in vitro* models suitable for each study.

The characteristic A/E lesions typically associated with an EHEC infection were not detected in this study, contrasting to previous reports using human colonoid-derived monolayers [2,13]. This discrepancy could be attributed to several factors. The use of a high-Wnt medium supplemented with fetal bovine serum (FBS) possibly suppresses the physiological differentiation of intestinal epithelial cells, leading to an immature cell population. A potentially important role of cellular differentiation in EHEC colonization has been proposed by documenting a significantly high colonization rate in differentiated human colonoid monolayers relative to undifferentiated counterparts [2]. Another explanation includes potentially insufficient bacterial exposure time prior to evaluation possibly limiting the establishment of bacterial-epithelial contact, the first step necessary for A/E lesion formation [23]. Additionally, the use of adult bovine organoids, as opposed to calf organoids, might reflect an inherent resistance of adult intestinal epithelium to EHEC as evidenced by the development of clinical disease in calf vs. largely subclinical carrier state in adult cattle [1,5]. Further studies under modified conditions are necessary to confirm *in vitro* formation of characteristic A/E lesions.

The study also observed greater mucus coverage of the apical surface in ileal monolayers relative to that observed in rectal monolayers, which is consistent with prior research [24]. This finding is not unexpected, given the ileum’s role in nutrient absorption in the presence of luminal content and its adaptive responses to microbial presence, which include bolstering physical barriers against pathogens and macromolecules. In contrast, a markedly low mucus coverage detected in rectal monolayers initially appears counterintuitive considering the rectum’s typically higher goblet cell density compared with the upper gastrointestinal tract [25]. However, this finding may partly be associated with differences in mucus composition, organization, and expression profile between the small and large intestine [26], possibly affecting the mucus preservation and visualization capability during the staining process. For instance, the mucus layer in small intestine is arranged in a single layer which is anchored to goblet cells, whereas that in large intestine consists of two layers: an outer nonattached and an inner attached layer [26]. The process of fixation for immunofluorescence examination might have led to the loss of some but not all mucus, particularly if the fixation technique is not optimized for preserving such structures. This technical aspect underscores the need for careful methodological choices in histological studies of mucus and suggests a potential area for improving our experimental design. Furthermore, the necessity for further differentiation of the rectal epithelial cells *in vitro* to achieve a physiologically representative mucus layer suggests that our model may not fully recapitulate the natural state of the rectal mucosa. This gap points to the complexity of replicating *in vivo* conditions within an *in vitro* setting, particularly for tissues like the rectum with specialized functions and cellular compositions.

The observed inter-segmental variations may offer insights into the pathophysiology of EHEC’s differential colonization and virulence within the bovine gastrointestinal tract. The study observed an increased mucus secretion in ileal monolayers following an infection with EHEC, which is protective of epithelial cells. In contrast, reduced mucus production was noted in the rectum under similar conditions. This decrease may represent a strategic vulnerability that EHEC exploits. Previous studies have indicated that an initial reduction or disruption of the mucus layer can occur during early infection stages, likely due to mucin-degrading enzymes such as zinc metalloprotease StcE produced by EHEC [9,27]. These changes were more pronounced in the rectum than in the ileum, with a notable increase in mucus production detected 14 days post-infection in an experimentally infected calf intestine [9], reflecting ongoing inflammatory stimuli. Concurrent with changes in mucus production, there was an upregulation of proinflammatory cytokine expressions such as *TNF-a* and *IL-8*. This upregulation is possibly an attempt by the host to eliminate or limit pathogen colonization [28]. The cytokine response contributes to the inflammatory environment, enhancing the host’s defensive measures against the bacterial invasion. EHEC’s ability to differentially colonize the ileum and rectum may be partially due to these varied host responses to infection. The pathogen’s success in colonizing the rectum might be facilitated by the reduced mucus layer and the extensive inflammatory response, which could provide a more favorable environment for bacterial persistence and activity. This adaptation is likely a critical factor in EHEC’s pathogenicity and its ability to establish infection and facilitate transmission within the host.

Utilizing SEM with a modified sample preparation technique, this research provided the first insight into the ultrastructure of bovine ileal and rectal mucus. While effective mucus preservation and visualization techniques have been explored extensively for histological and immunostaining evaluations [29,30,31,32], comparative studies for ultrastructural evaluations with SEM remain sparse [16]. The modified GA/PFA/AB-fixation technique explored in this study effectively preserved and visualized the fine net-like structure of mucus in our bovine model, which aligns with the structures seen in human and porcine intestinal mucus [16]. On the other hand, a conventional GA-fixation technique preserved little mucus and effectively visualized the underlying microvilli, another important target to be evaluated with electron microscopy. The study highlights the critical importance of selecting suitable fixation techniques for visualizing distinct luminal structures like microvilli and mucus. This choice of sample preparation methods is essential, depending on the specific objectives of the research.

The current study has several limitations that warrant mention. First, the use of FBS-supplemented high-Wnt culture medium, although effective in maintaining stable monolayers and facilitating EHEC infection, may not support full physiological differentiation of cells [33]. Additionally, the relatively short exposure of the monolayers to EHEC does not fully mimic the *in vivo* intestinal colonization by EHEC. Prolonging the co-culture period was considered but dismissed due to the risk of bacterial overgrowth, which could lead to exaggerated and potentially misleading outcomes in our static system. To address these issues, future research could benefit from employing a more dynamic system, such as a microfluidic gut-on-a-chip model, which has shown promise in human studies [34,35].

## 4. Materials and Methods

### 4.1. Crypt Isolation and Organoid Culture from Adult Bovine Ileal and Rectal Tissue

Adult bovine intestinal tissue was sampled, processed and cultured for organoid generation according to a previous study [36]. Briefly, sections of ileum and rectum were grossly identified and approximately 10 to 15 pieces of fresh tissue were collected using biopsy forceps. All samples were obtained from 15- to 18-month-old cattle at a local slaughterhouse. The tissue samples were placed in ice-cooled wash medium which consisted of 1× penicillin/streptomycin and 25 μg/mL gentamicin in Dulbecco’s phosphate-buffered saline (PBS) and transported to the lab for further processing. Following a vigorous rinsing with the wash medium, stem cell-containing intestinal crypts were isolated from the tissue by mincing and incubating in a 20 mM ethylenediaminetetraacetic acid (EDTA) solution at 4 °C for 15 (ileum) or 60 (rectum) minutes on a tube rotator. The supernatant was collected and centrifuged at 200× *g* at 4 °C for 5 min to pellet the crypts, which was then resuspended in Matrigel and seeded onto a 48-well plate with 30 μL per well. Following polymerization of Matrigel at 37 °C for 10 min, 300 μL of organoid culture medium was added to each well. The medium was changed every other day. Growing organoids were subcultured every 6–8 days by recovering them from Matrigel by incubating in Cell Recovery Solution at 4 °C for 60 min, dissociating with TrypLE Express at 37 °C for 1 min, and reseeding in Matrigel at an expansion ratio of approximately 1:6 as described previously [36]. Organoid culture medium was prepared according to a previous study [36] and its composition is summarized in Appendix A.

### 4.2. Organoid-Derived Monolayer Culture

Adult bovine ileal and rectal organoid-derived monolayers were generated according to previously described protocols with some modifications [37,38]. Briefly, ileal and rectal organoids that were cultured and expanded as above were recovered from Matrigel, dissociated to single cells with TrypLE Express, supplemented with 10 μM Y-27632 at 37 °C for 10 min, filtered through a 70 μm cell strainer, and resuspended to a concentration of 2.5 × 10^6^ cells/mL for ileal and 1.5 × 10^6^ cells/mL for rectal cells in their respective monolayer culture media (Appendix A). Subsequently, 200 μL of the cell suspension was seeded onto a 24-well cell culture insert which was precoated with 2% Matrigel in Advanced DMEM/F12, supplemented with 2 mM GlutaMAX-I and 10 mM HEPES at 37 °C for 1 h. The inserts prepared for the rectum were further incubated at 37 °C overnight in monolayer culture medium prior to cell seeding. Finally, 500 μL of respective monolayer culture media was added to the basolateral chamber of each well. The medium was changed every other day until stable monolayers were formed.

### 4.3. Evaluation of Epithelial Barrier Integrity

Stability of the monolayers was assessed using phase contrast microscopy (DMi1, Leica), TEER measurements, and *P*_app_ assays. The details of each technique have been described elsewhere [39,40]. Briefly, the TEER (Ω*cm^2^) was measured daily using epithelial Volt-Ohm Meter (Millicell ERS-2, Millipore AG, Burlington, MA, USA), subtracting the blank value measured in Ω and multiplying by the cell culture surface area (0.33 cm^2^) [37]. The *P*_app_ (cm/s) of 4 kDa FITC-dextran was determined on day 1, 3, and 5 of culture by measuring the fluorescence intensity of the basolateral culture medium over 100 min following apical application of 0.5 mg/mL FITC-dextran to each well. The measurements were obtained with excitation and emission wavelengths of 495 and 535 nm, respectively, using a SpectraMax i3x microplate reader (Molecular Devices) [37]. The monolayers were considered stable once cells grew to confluence and TEER and *P*_app_ values reached a plateau. At least two technical replicates per experiment and more than three biological replicates were assessed to ensure the repeatability. Subsequent experiments were conducted using monolayers which had reached a stable state.

### 4.4. EHEC Culture and Monolayer Infection

A wild strain of *E. coli* O157:H7 isolated from cattle feces and tested positive for major virulence genes, namely *EAE*, *STX1*, *STX2/VT2* and *H7*, with PCR analysis was used in this study. An overnight culture of EHEC was prepared by incubating in 3 mL of Luria-Bertani (LB) broth at 37 °C with shaking at 200 rpm, which was then diluted 1:10 into fresh LB broth and cultured further for 1.5 h to a late log phase (Appendix A). Bacteria were harvested, washed with PBS, and resuspended into a concentration of 1 × 10^8^ CFU/mL in the monolayer culture medium. Subsequently, 10 μL of the bacterial suspension was inoculated to the apical chamber of monolayers and incubated at 37 °C for 4 h. The 4 h incubation period was chosen to prevent complications from bacterial overgrowth and changes in culture medium pH observed after this period, ensuring clearer interpretation of host–pathogen interactions. Additionally, an appropriate control without EHEC exposure, i.e., the monolayers which were mock infected with the monolayer culture medium only, was included to rule out the influence of any external variables on the results. At least two technical replicates per condition were prepared in each experiment.

### 4.5. Electron Microscopy

To assess host–bacterial interactions and the impact of EHEC on the mucus at 4 h post infection, ileal and rectal monolayers were fixed and processed for SEM and TEM imaging as described previously with minor modifications [37,38]. For SEM, monolayers were fixed either with 2.5% glutaraldehyde in a 0.1 M sodium cacodylate buffer (GA) or with 2% glutaraldehyde mixed with 2% paraformaldehyde and 1.05% alcian blue in a 0.1 M cacodylate buffer (GA/PFA/AB) overnight at 4 °C. Following 0.1 M cacodylate buffer rinses, GA-fixed samples were post-fixed with 1% osmium tetroxide for 2 h at 4 °C. GA/PFA/AB-fixed samples were post-fixed with 1% osmium tetroxide and 0.5% tannic acid in a 0.1 M cacodylate buffer under the same condition. Subsequently, both samples were dehydrated with a graded ethanol series and finally in hexamethyldisilazane (HMDS). Finally, the cell culture membrane was cut out from the insert and mounted on a carbon conducting tab, coated with Pt/Pd sputter coater (Cressington High Resolution Sputter Coater) and imaged with a Quanta 200F SEM (FEI).

For TEM, monolayers were fixed with GA/PFA/AB and rinsed with a 0.1 M cacodylate buffer as above. The samples were post-fixed with 1% osmium tetroxide and 1% ferrocyanide in a 0.1 M cacodylate buffer for 2 h and 2% uranyl acetate for 1 h at room temperature. Subsequently, the samples were dehydrated with a graded ethanol series and finally in propylene oxide, infiltrated with a 1:1 Spurrs resin and propylene oxide mix overnight, and embedded in 100% Spurrs resin, which was polymerized in an oven at 60 °C for two days. Finally, the samples were sectioned to 80 nm thickness, stained with uranyl acetate, potassium permanganate and Raynold’s lead, and imaged with a Tecnai G2 20 Twin TEM (FEI).

The number of adhering bacteria observed with SEM were quantified on randomly selected representative images captured under low (1542×) magnification. Five independent fields of view from three biological replicates were evaluated. The number of bacteria was normalized by the surface area of cell culture insert (0.33 cm^2^) and expressed relative to the number of bacteria in the inoculum.

### 4.6. RT-qPCR

RT-qPCR was performed to evaluate changes in expression levels of genes for proinflammatory cytokines (*TNF-a*, *IL-6*, and *IL-8*), goblet cells (*MUC2*), actin filament (*F-actin*), and tight junction proteins (*CLDN1*, *OCLN*, *TJP1*, and *TJP2*) according to a previous study [36]. Briefly, total RNA was extracted from control and EHEC-infected monolayers at 4 h post infection using an RNeasy Plus Mini Kit (Qiagen, Germantown, MD, USA) and transcribed to cDNA using a High Capacity cDNA Reverse Transcription Kit (Applied Biosystems, Foster City, CA, USA) following the manufacture’s protocols. RT-qPCR was performed using a PowerUp SYBR Green Master Mix (Applied Biosystems), with the primer sequence being amplified at 60 °C for 40 cycles. Relative expression levels of target genes were determined by using *GAPDH*, *RPL0*, and *ACTB* as the internal control [41,42,43] and compared between control and EHEC-infected monolayers. Primers used in this study were adopted from previous studies and summarized in Table 1 [42,43,44,45]. RT-qPCR reactions were carried out in duplicate from three biological replicates with two technical replicates per experiment.

### 4.7. Immunocytochemistry

The impact of EHEC infection on the structural integrity and the mucus of ileal and rectal monolayers was assessed with confocal microscopy following immunofluorescence staining, which was performed according to a previous study [36]. Briefly, control and EHEC-infected monolayers were treated with 4% paraformaldehyde (15 min), 0.3% Triton X-100 (10 min), and 2% bovine serum albumin (BSA) (60 min) at room temperature for fixation, permeabilization, and blocking, respectively. The samples were incubated with Alexa Fluor 488-conjugated primary antibody against E-cadherin (BD Biosciences, Franklin Lakes, NJ, USA, 1:200) or fluorescein-conjugated SNA probe (Vector Laboratories, 1:100, Newark, CA, USA) diluted in 2% BSA overnight at 4 °C in the dark for the visualization of adherens junction protein and mucus, respectively. Subsequently, all samples were stained for F-actin (Alexa Fluor 647-conjugated phalloidin, Invitrogen, 1:400) and nuclei (4′,6-diamidino-2-phenylindole dihydrochloride (DAPI), Thermo Scientific, 1:1000, Waltham, MA, USA) following manufacturer’s recommendations before the membrane was cut out from cell culture inserts and mounted on glass slides using a ProLong Gold Antifade reagent. A white-light point scanning confocal microscope (SP8-X, Leica Microsystems, Deerfield, IL, USA) and LAS X (Leica Microsystems, Deerfield, IL, USA) were used to capture and process immunofluorescence images.

Percent positive area stained with SNA per unit area was determined using ImageJ 1.54h software (National Institutes of Health, Bethesda, MD, USA) on randomly selected representative images which were captured under high (63×) magnification. At least three independent fields of view per slide from three independent experiments were evaluated using at least three biological replicates.

### 4.8. Statistical Analyses

Quantitative data were analyzed using R v.3.4.1 (The R foundation) and plotted using GraphPad Prism 9.5.1 (GraphPad Software). Data were compared between ileal and rectal monolayers using Student’s *t*-test or between control and EHEC-infected monolayers using Wilcoxon’s signed rank tests. The results were presented as mean ± standard error of the mean (sem), with the significance level being set at *p* < 0.05.

## 5. Conclusions

This study demonstrated the utility of adult bovine ileal and rectal organoid-derived monolayers as effective *in vitro* models for studying EHEC colonization. These models successfully replicated key interactions between EHEC and bovine tissues, including bacterial adherence and immune responses, highlighting segment-specific differences in susceptibility and inflammation. The lack of A/E lesions and variable mucus coverage emphasize the need for model optimization to mirror *in vivo* conditions more closely. Nonetheless, the future applications of these models are broad, including investigations in segment- or species-specific molecular mechanisms regulating EHEC infections in cattle. The approach can also be applied to other enteric pathogens of veterinary and public health importance, as well as commensal microbiota to study host–pathogen–microbiota interactions. Our findings advance the understanding of EHEC pathogenesis and underline the importance of tailored *in vitro* models for developing strategies to prevent EHEC transmission, enhancing public health protection.

## Figures and Tables

**Figure 1 ijms-25-04914-f001:**
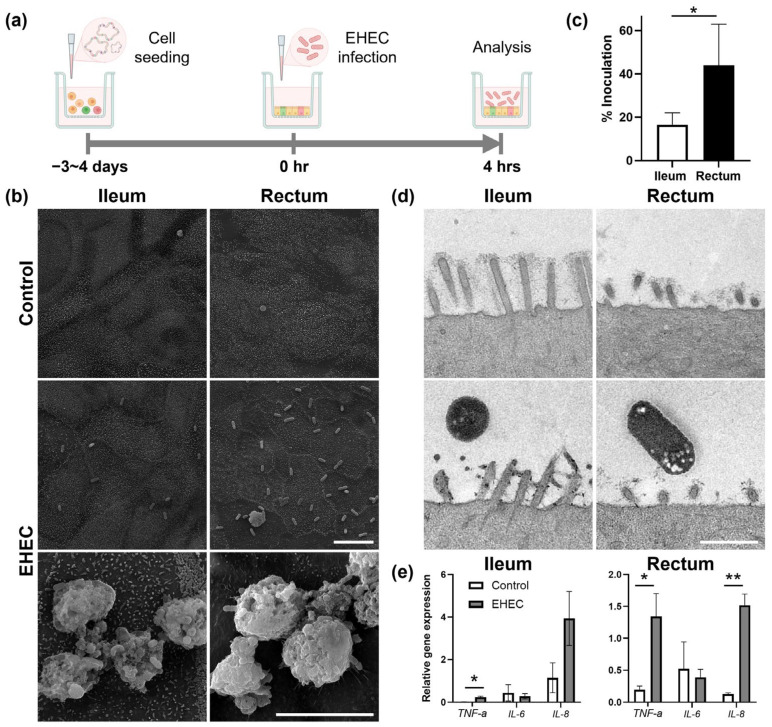
Infection of bovine ileal and rectal organoid-derived monolayers with EHEC. (**a**) Schematic depicting experimental timeline from generation of monolayers to infection with EHEC. Three-dimensional organoids were dissociated into single cells and seeded on to a cell culture insert to generate a confluent monolayer approximately 3–4 days prior to the experiment. A stable monolayer was infected with EHEC at time 0 and co-cultured for 4 h prior to analyses. Created with BioRender.com. (**b**) Representative scanning electron microscopy (SEM) images of control and EHEC-infected monolayers demonstrated bacterial adherence to the apical surface of the epithelium and extruded mucus. Scale bars, 10 μm. (**c**) Quantification of adhering bacteria on the apical surface of the monolayers observed in SEM images revealed preferential colonization of EHEC to the rectum over the ileum. Five independent fields of view from three independent experiments using three biological replicates were analyzed. Data are expressed as the mean ± sem. * *p* < 0.05. (**d**) Representative transmission electron microscopy (TEM) images of control and EHEC-infected monolayers demonstrated perturbation and effacement of microvilli in ileal and rectal monolayers upon infection with EHEC, respectively. Scale bar, 1 μm. (**e**) RT-qPCR of control and EHEC-infected monolayers documented upregulation of *TNF-a* and *IL-8*, but not *IL-6*, genes in EHEC-infected monolayers at 4 h. Relative gene expression levels were calculated using *GAPDH*, *RPL0*, and *ACTB* as the internal control. Two technical replicates from three biological replicates were evaluated. Data are expressed as the mean ± sem. * *p* < 0.05, ** *p* < 0.01.

**Figure 2 ijms-25-04914-f002:**
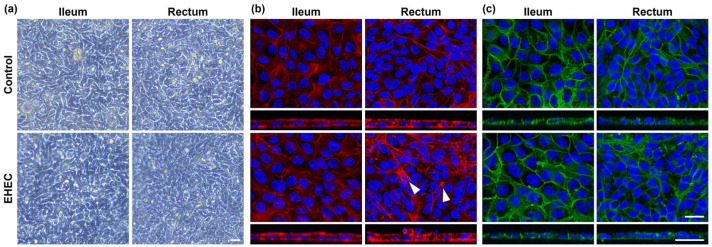
Impact of EHEC infections on the integrity of bovine ileal and rectal organoid-derived monolayers. (**a**) Representative phase contrast images of control and EHEC-infected monolayers demonstrating intact monolayers in both ileal and rectal monolayers at 4 h post infection. Scale bar, 20 μm. (**b**,**c**) Representative immunofluorescence images of control and EHEC-infected monolayers. While apical membrane perturbation and actin rearrangement (arrowheads) are observed in EHEC-infected monolayers (**b**), little difference was noted in the distribution of adherens junction or E-cadherin between the groups (**c**). Top-down views (top) and z-stack images (bottom) are shown. F-actin: red; E-cadherin: green; nuclei: DAPI, blue. Scale bars, 20 μm.

**Figure 3 ijms-25-04914-f003:**
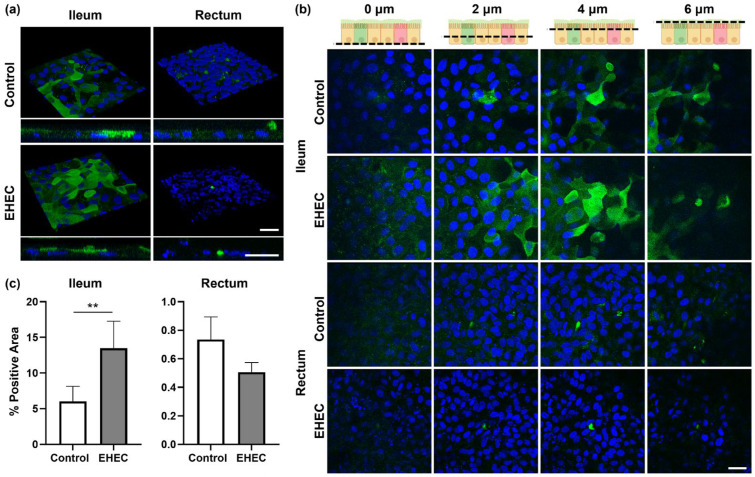
Impact of EHEC infection on mucus secretion in bovine ileal and rectal monolayers. (**a**) Representative 3D reconstruction and z-stack images of control and EHEC-infected monolayers visualizing mucus with immunofluorescence staining against SNA. Scale bars, 20 μm. (**b**) Top-down views of z-stack images captured at 2 μm intervals from the bottom of the monolayers to the top are shown. Schematics illustrate relative locations of z-stack images presented in each column. Created with BioRender.com. SNA: green; nuclei: DAPI, blue. Scale bar, 20 μm. (**c**) Percent positive area stained with SNA per unit area was compared between control and EHEC-infected monolayers. Three independent fields of view from three independent experiments using three biological replicates were analyzed. Data are expressed as the mean ± sem. ** *p* < 0.01.

**Figure 4 ijms-25-04914-f004:**
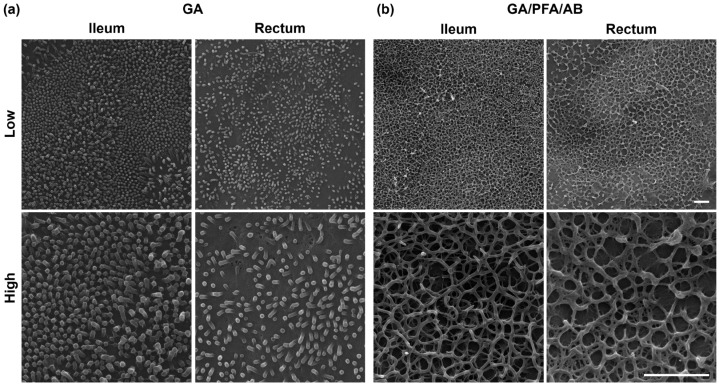
Comparison of two fixation techniques for preservation and visualization of mucus ultrastructure. Representative scanning electron microscopy images of control ileal and rectal monolayers captured with lower (top) and higher (bottom) magnifications are shown. (**a**) GA-fixation preserved little mucus and clearly depicted apical surface cellular structure. Microvilli were more densely packed and elongated in ileal monolayers compared with rectal monolayers. (**b**) GA/PFA/AB-fixation preserved more surface covering mucus, thus allowed visualization of mucus ultrastructure, which demonstrated uniform net-like 3D structure in both ileal and rectal monolayers. Scale bars, 2 μm.

**Figure 5 ijms-25-04914-f005:**
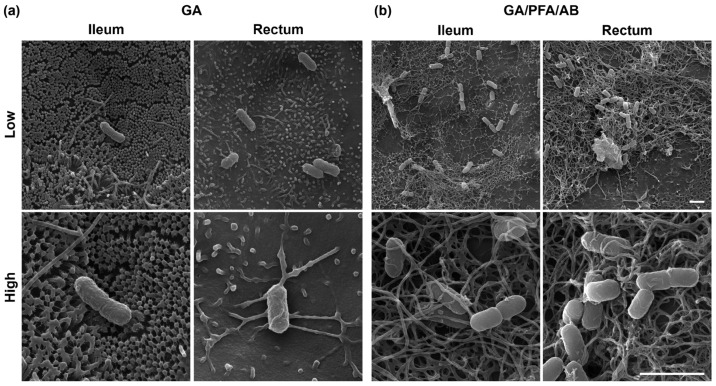
Impact of EHEC infection on microvilli and mucus structures in bovine ileal and rectal monolayers. Representative scanning electron microscopy images of EHEC-infected ileal and rectal monolayers captured with lower (top) and higher (bottom) magnifications are shown. (**a**) Microvilli effacement and perturbation was evident in rectal monolayers but less pronounced in ileal monolayers fixed with GA. (**b**) Net-like structure of mucus was disturbed and disorganized in both ileal and rectal monolayers. Adhering bacteria were closely associated with elongated mucus strands. The change was more pronounced in rectal than in ileal monolayers. Scale bars, 2 μm.

**Table 1 ijms-25-04914-t001:** Primers used for RT-qPCR analysis of bovine organoid-derived monolayers.

Gene	Gene Name	Forward	Reverse	Reference
*ACTB*	β-actin	CTAGGCACCAGGGCGTAATG	CCACACGGAGCTCGTTGTAG	[41]
*CLDN1*	*Claudin 1*	*TTCGACTCCTTGCTGAATCTG*	*GGCTATTAGTCCCAGCAGGATG*	[42]
*F-actin*	*F-actin*	*AATCAGAGGCCAAGGGAACT*	*TGCAGGATGAGCTTGTTGTC*	[44]
*GAPDH*	Glyceraldehyde-3-phosphate dehydrogenase	ATCTCGCTCCTGGAAGATG	TCGGAGTGAACGGATTCG	[43]
*IL-6*	Interleukin 6	ACCCCAGGCAGACTACTTCT	GCAAATCGCCTGATTGAACCC	[42]
*IL-8*	Interleukin 8	TGCTTTTTTGTTTTCGGTTTTTG	AACAGGCACTCGGGAATCCT	[45]
*MUC2*	Mucin 2	TTCGACGGGAGGAAGTACAC	TTCACCGTCTGCTCATTCAG	[43]
*OCLN*	*Occludin*	*CCTTTTGAAAGTCCACCTCCTTAT*	*TGTCATTGCTTGGTGTGTAGT*	[42]
*RPL0*	Ribosomal protein L0	CAACCCTGAAGTGCTTGACAT	AGGCAGATGGATCAGCCA	[42]
*TJP1*	*Tight junction protein 1*	*AGAAAGATGTTTATCGTCGCATCGT*	*ATTCCTTCTCATATTCAAAATGGGTTCTGA*	[42]
*TJP2*	*Tight junction protein 2*	*TGCTCCATTCATTTGCGGTTC*	*GGCCTCTTGACCACAATGG*	[42]
*TNF-a*	Tumor necrosis factor alpha	AGAGGGAAGAGCAGTCCCCAG	TTCACACCGTTGGCCATGAG	[42]

## Data Availability

All data associated with this study are included in the article and Appendix A.

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
