# Peer review of "Differential Colonization and Mucus Ultrastructure Visualization in Bovine Ileal and Rectal Organoid-Derived Monolayers Exposed to Enterohemorrhagic Escherichia coli"

_ijms, 2024, doi:10.3390/ijms25094914_

Round 1

Reviewer 1 Report

Comments and Suggestions for Authors

Authors conducted a study to explore the promising approach for studying EHEC colonization using bovine organoid-derived monolayers. However, revisions are required in following areas to improve its quality.

Major Suggestion:

Abstract:

1. Authors are suggested to improve the abstract particularly regarding the methods employed and the results obtained. It would be beneficial to include quantitative findings observed in the study e.g. the extent of bacterial adherence, changes in cytokine expression levels and differences in mucus coverage between ileal and rectal monolayers.

2. Additionally, Authors may mention any limitations of the research and potential implications for future studies.

Introduction:

3. Authors are suggested to provide more comprehensive background information on previous studies related to EHEC colonization in cattle and the limitations of existing in vitro models.

Materials and Methods:

.4. Authors should elaborate the rationale behind using adult bovine ileal and rectal organoid-derived monolayers as in vitro models and the choice of specific culture conditions and media components.

5. In Addition, it would be helpful to include quality control measures undertaken to ensure the reliability of the experimental procedures.

Discussion:

6. Authors are suggested to compare and contrast the observed results with previous studies. Additionally, authors are suggested to address the significance of inter-segmental variations in mucus coverage and cytokine expression, regarding EHEC colonization and infection dynamics.

7. Authors should discuss their finding to explore the implications in the context of EHEC pathogenesis, host-pathogen interactions and the development of preventive strategies.

8. Authors are suggested to discuss the limitations as well.

Conclusion:

9. Authors are suggested to include the valuable recommendations for future research directions and potential applications in veterinary medicine.

Reviewer 2 Report

Comments and Suggestions for Authors

In this manuscript the authors study the enterohemorrhagic Escherichia coli using advanced organoid-derived monolayers from adult bovine ileum and rectum in order to investigate the intricacies of enterohemorrhagic Escherichia coli colonization, the differential responses of intestinal tissues, and the protective functions of the mucus barrier. The work is of high importance since enhances the understanding of enterohemorrhagic Escherichia coli pathogenesis in cattle. 

I recommend the manuscript to be considered for publication after minor revision. 

L 80-83: regarding tissue tropism of EHEC colonization, this part refers to the results obtained in other works, therefore it would be more appropriate for this phrase to be included in the discussion chapter.

L93-95 and L 120-121: the same observation as for L80-83.

L313: Please do not leave space between the value and Celsius degrees (°C). Please be consistent throughout the manuscript.

Regarding In the citations, from the total of 36 citation, 3 are self-citations by authors.

Reviewer 3 Report

Comments and Suggestions for Authors

This manuscript by Minae Kawasaki et al. reveals organoid monolayers derived from the rectum and ileum of cattle to demonstrate their usefulness as models of EHEC infection. The objectives, methods, results, and discussion in this manuscript are adequately described. Some minor concerns are as follows;

•The authors should clarify the strain name of the EHEC strain isolated from cattle used in this study. It should also be mentioned that the authors used wild strains of bovine origin rather than typical strains.

•Figure 1c: P value (*) is missing.

•Line 142: It is necessary to explain for what purpose the MUC2 gene was measured.

•Line 327: Figure S1A -> S2A

•Line 330: Figure S1B -> S2B

•Line 362: Add a description of MUC2 to the RT-qPCR section.

•Table 1: Add primer sequence of MUC2.
